# A Likelihood-Based Pose Estimation Method for Robotic Arm Repeatability Measurement Using Monocular Vision

**DOI:** 10.3390/s25227089

**Published:** 2025-11-20

**Authors:** Peng Zhang, Jiatian Li, Jiayin Liu, Feng He, Yiheng Jiang

**Affiliations:** Faculty of Land and Resources Engineering, Kunming University of Science and Technology, Kunming 650093, China; 20232101047@stu.kust.edu.cn (P.Z.); 20233101016@stu.kust.edu.cn (J.L.); 20222201097@stu.kust.edu.cn (F.H.); rayjj@stu.kust.edu.cn (Y.J.)

**Keywords:** repeatability, maximum likelihood estimation, pose estimation, Cramér–Rao bound, Iterative Closest Point (ICP)

## Abstract

Repeatability accuracy is a key performance metric for robotic arms. To address limitations in existing monocular vision-based measurement methods, this study proposes a likelihood-based pose estimation approach. Our method first obtains initial pose estimates through optimized likelihood estimation, then iteratively refines depth information. By modeling the statistical characteristics of multiple observed poses, we derive a global theoretical pose. Within this framework, two-dimensional feature points are backprojected into three-dimensional space to form an observed point cloud. The error between this observed cloud and the theoretical feature point cloud is computed using the Iterative Closest Point (ICP) algorithm, enabling accurate quantification of repeatability accuracy. Based on 30 repeated trials at each of five target poses, the proposed method achieved repeatability positioning accuracy of 0.0115 mm, 0.0121 mm, 0.0068 mm, 0.0162 mm, and 0.0175 mm at the five poses, respectively, with a mean value of 0.0128 mm and a standard deviation of 0.0038 mm across the poses. Compared with two existing monocular vision-based methods, it demonstrates superior accuracy and stability, achieving average accuracy improvements of 0.79 mm and 1.06 mm, respectively, and reducing the standard deviation by over 85%.

## 1. Introduction

In the field of intelligent manufacturing and automated production, repeatability is a core metric for evaluating the performance of robotic arms and other equipment. Its measurement accuracy directly impacts product quality and production efficiency [1,2,3]. High-precision repeatability measurement is fundamental to ensuring the reliability of processes such as assembly, welding, and inspection.

Currently, robotic arm repeatability measurement methods are categorized into laser-based and vision-based approaches according to the sensor type. Laser measurement techniques [4,5,6,7] capture coordinate data by detecting reflected laser spots on target surfaces, demonstrating significant advantages in specific scenarios due to their high measurement accuracy. However, this approach suffers from limitations, including high equipment costs, poor environmental adaptability, and the requirement for specialized targets. The complex installation and calibration procedures make it difficult to meet the practical demands of complex and dynamic industrial environments. In contrast, visual measurement methods offer superior scene adaptability, more convenient deployment methods, and lower cost investment. Within the visual measurement technology framework, pose estimation constitutes a critical component for achieving high-precision measurement. Diverse pose estimation methods provide multiple technical pathways for visual repeatability accuracy measurement. Current research primarily focuses on three major directions: camera calibration methods, non-iterative methods, and iterative methods. Camera calibration methods establish a transformation relationship between the image coordinate system and the world coordinate system to achieve accurate measurement of the end-effector pose of a robotic arm. R.A. Boby et al. [8] proposed a phased monocular camera calibration method and applied it to robotic arm repeatability measurement tasks. Wu et al. [9] introduced a vision-based robotic motion measurement approach, enhancing measurement accuracy and stability through monocular feature matrices and singular value decomposition. Non-iterative methods rely on analytical geometric constraints, achieving measurement through direct pose parameter solutions. To enhance algorithmic efficiency and accuracy, ref. [10] proposed the EPnP model, and multiple improved variants were subsequently derived, including RPnP [11], DLS [12], OPnP [13], SRPnP [14], and the deep learning-based EPro-PnP model [15]. However, measuring robotic repeatability accuracy under monocular vision represents a unique task paradigm. Its core objective involves performing multiple observations of the same scene in a static state and evaluating their consistency. This task focuses more on maximizing the measurement accuracy and stability of single-frame images without relying on temporal sequence information. Consequently, the widely recognized baseline methods remain the classical PnP algorithms and their improved variants. Among these, Gao et al. [16] proposed an image-based target localization method for collaborative robots, utilizing a monocular vision system combined with a PnP algorithm to measure robotic arm repeatability accuracy. Luo et al. [17] employed a coplanar P4P algorithm to determine target feature point coordinates in the camera coordinate system, integrating dual quaternions to solve the pose transformation matrix and thereby calculating repeatability accuracy. The iterative method transforms the PnP problem into an optimization objective function, solving the pose through iterative steps. Chen et al. [18] proposed a robotic arm positioning measurement system combining monocular vision with P4P and OI algorithms [19] that employs P4P results as initial values for OI algorithm optimization. However, most existing visual measurement methods scarcely consider the statistical properties of observational data and lack global optimization mechanisms for multi-frame data.

To address this, we propose a likelihood-based pose method for achieving high-precision repeatability measurement. This method first optimizes the estimated single-frame pose, then employs a likelihood-based pose estimation framework: based on maximum likelihood estimation theory, it transforms the pose problem of solving PnP for a stationary target observed multiple times into a problem of observing a slightly moving target multiple times through random signal modeling. Subsequently, the Iterative Closest Point (ICP) algorithm [20] computes displacement vectors between 3D–3D matched point pairs. The minimum envelope sphere radius is then determined using the progressive algorithm with a preference for the farthest point. This enables precise quantification of repeatability accuracy, overcoming the limitation of traditional single-frame measurement methods that cannot perform global optimization. Furthermore, by calculating 3D–3D point cloud deviations, more accurate and stable repeatability accuracy is obtained.

## 2. Materials and Methods

Repeatability accuracy measures the degree of pose dispersion across multiple responses to identical commands, with its core challenge lying in quantifying end-effector pose repeatability. Based on the aforementioned theory, this study employs the eye-in-hand model. By establishing a rigid mechanical connection between the visual sensor and the end-effector, a fixed coordinate system transformation relationship is constructed. Subsequently, by calculating the repeatability of the visual sensor’s pose, an indirect measurement of the end-effector’s repeatability accuracy is achieved. The overall process is illustrated in Figure 1.

The overall process comprises three modules: data acquisition, pose estimation, and repeatability accuracy calculation. The data acquisition module utilizes an industrial camera to capture image data, establishing the foundational geometric data for subsequent computations. The pose estimation module employs a weighted likelihood method to refine the initial pose estimates from PnP, mitigating the impact of observation outliers caused by image noise. Based on the optimized pose, a two-stage iterative algorithm solves single-frame depth information, explicitly enhancing estimation stability in the depth direction; by constructing a random signal model and employing maximum likelihood theory, it fuses multi-frame optimized poses to derive a globally optimal pose solution as the theoretical pose; Within the theoretical pose framework, depth information that is iteratively optimized is utilized to backproject two-dimensional feature points onto the world coordinate system, forming a three-dimensional observation point cloud under theoretical pose observation. The ICP algorithm is then employed to compute the registration error between the observation point cloud and the theoretical three-dimensional feature point cloud. The repeatability accuracy calculation module determines the final repeatability accuracy result by computing the minimum envelope sphere radius of multi-frame registration errors.

### 2.1. Initial Pose Estimation and Optimization

Within the data acquisition module, image data from a calibration plate is captured using an industrial camera. The calibration plate image acquisition protocol adheres to the repeatability accuracy data collection method specified in the national standard ISO 9283 [21]. Position the calibration plate within the workspace of the robotic arm under test. Using the teach pendant, program the control sequence to write five distinct poses into the controller, designated as P1, P2, P3, P4, and P5, respectively. Command the robotic arm to traverse repeatedly between two points at normal operating speed, capturing image data upon reaching each target pose. Each image acquisition sequence is repeated 30 times. The detailed image acquisition workflow is illustrated in Figure 2.

To convert abstract image information into quantifiable pose data, the PnP algorithm is employed for initial camera pose estimation. Processing images acquired from a single observation yields n pairs of 2D–3D matched feature points. For the *k*-th observation, the coordinates of the *i*-th feature point in the world coordinate system are denoted as $P_{i}^{w}\left(k\right)={\left[X_{i}^{w}\left(k\right),Y_{i}^{w}\left(k\right),Z_{i}^{w}\left(k\right)\right]}^{T}$, and its pixel coordinates in the image coordinate system are $P_{i}^{img}\left(k\right)={\left[u_{i}^{k},v_{i}^{k}\right]}^{T}$, where *i* = 1, 2,..., *n*. The transformation relationship can be expressed using the ideal pinhole camera model:(1)$$\left\{\begin{array}{c} u_{i}^{k}=f_{x}\frac{r_{1}^{k}P_{i}^{w}\left(k\right)+t_{x}^{k}}{r_{3}^{k}P_{i}^{w}\left(k\right)+t_{z}^{k}}+c_{x}\\ v_{i}^{k}=f_{y}\frac{r_{2}^{k}P_{i}^{w}\left(k\right)+t_{y}^{k}}{r_{3}^{k}P_{i}^{w}\left(k\right)+t_{z}^{k}}+c_{y} \end{array}\right.$$

$Cam=\{f_{x},f_{y},c_{x},c_{y}\}$ denotes the camera intrinsic parameters, obtainable through calibration experiments and representing fixed values. To quantify the precision of the calibration, we report the maximum re-projection error across all calibration images, which is 0.11 pixels. This low value confirms the accuracy and reliability of the estimated intrinsic parameters $Cam=\{f_{x},f_{y},c_{x},c_{y}\}$ for our subsequent visual odometry computations. $t^{k}={\left[t_{x}^{k},t_{y}^{k},t_{z}^{k}\right]}^{T}$ is the translation vector, and $R^{k}={\left[r_{1}^{k},r_{2}^{k},r_{3}^{k}\right]}^{T}$ denotes the rotation matrix, where *k* = 1, 2,..., *K*, and $t^{k}$ and $R^{k}$ are initial pose parameters to be estimated, characterizing the camera’s pose during the *k*-th observation.

As images in practical applications often contain noise, the two-dimensional feature point $P_{i}^{img}\left(k\right)$ observed at the *k*-th observation is typically accompanied by observation noise ϵ(k). Since the initial pose estimation is based on the PnP algorithm, which is sensitive to noise in two-dimensional feature points, this noise introduces bias into the estimated initial pose $t^{k}$ and $R^{k}$, thereby affecting subsequent theoretical pose estimation. To safeguard the accuracy of subsequent theoretical pose estimation, nonlinear optimization was performed on the initial pose $t^{k}$ and $R^{k}$. Currently, many nonlinear optimization-based VO systems explicitly assume that visual feature measurement noise is Gaussian white noise [22,23]. Based on this, it is assumed that at the *k*-th observation, the observation noise ϵ(k) follows a Gaussian distribution with mean zero and variance $\sigma^{2}$, and its projection process can be expressed as(2)$$P_{i}^{img}\left(k\right)=\pi \left(R^{k},t^{k},P_{i}^{w}\left(k\right)\right)+\epsilon \left(k\right)$$
where $\pi (R^{k},t^{k},P_{i}^{w}\left(k\right))$ denotes the projection function of the ideal pinhole camera model. At the *k*-th observation, to account for uncertainty in feature point extraction, a weight $w\left(k\right)={\left[w_{1}^{k},w_{2}^{k},\ldots ,w_{n}^{k}\right]}^{T}$ is introduced for each 2D feature point to mitigate the impact of outliers on pose estimation results. This paper employs an improved IGG 3 function to determine the weights, which exhibits favorable robustness. The weight of the *i*-th feature point can be expressed as(3)$$w_{i}^{k}=\left\{\begin{array}{c} 1\;\left|e_{i}\right|\leq k_{0}\\ \frac{k_{0}}{\left|e_{i}\right|}{\left(\frac{k_{1}-\left|e_{i}\right|}{k_{1}-k_{0}}\right)}^{2}\;k_{0}<\left|e_{i}\right|<k_{1}\\ 0\;\left|e_{i}\right|\geq k_{1} \end{array}\right.$$
where $k_{0}$ and $k_{1}$ are harmonic coefficients, and $e_{i}$ is the standardized residual:(4)$$\left\{\begin{array}{c} e_{i}=\frac{e_{i}^{\prime }}{\sigma_{0}}\\ \sigma_{0}=1.4826median\left(e_{i}^{\prime }\right) \end{array}\right.$$
where $e_{i}$ is the reprojection error for each 2D feature point, $e_{0}$ is the normalization factor, and $median(e_{i}^{\prime })$ is the median of $e_{i}$.

As the two-dimensional feature points are mutually independent, the joint probability density for *n* such points is expressed as in Equation (Equation 5). Taking the natural logarithm of Equation (Equation 5) yields the likelihood function, as shown in Equation (Equation 6). From Equation (Equation 6), it follows that maximizing the likelihood function is equivalent to minimizing the following function, thereby enabling the solution of the optimized camera pose, as depicted in Equation (Equation 7).(5)$$P=\prod_{i=1}^{n}P\left(P_{i}^{img}\left(k\right)|\pi \left(R^{k},t^{k},P_{i}^{w}\left(k\right)\right),w\left(k\right),\sigma^{2}\right)$$(6)$$\begin{array}{cc} \; & lnL(P_{i}^{img}\left(k\right)|\pi \left(R^{k},t^{k},P_{i}^{w}\left(k\right)\right),w\left(k\right),\sigma^{2})\\ \; & =-\sum_{i=1}^{n}w\left(k\right)ln\left(2\pi \sigma^{2}\right)-\frac{1}{\sigma^{2}}\sum_{i=1}^{n}w\left(k\right){\left(P_{i}^{img}\left(k\right)-\pi \left(R^{k},t^{k},P_{i}^{w}\left(k\right)\right)\right)}^{2} \end{array}$$(7)$$\left[{\tilde{R}}^{k},{\tilde{t}}^{k}\right]=\min\left\{\sum_{i=1}^{n}w\left(k\right){\left(P_{i}^{img}\left(k\right)-\pi \left(R^{k},t^{k},P_{i}^{w}\left(k\right)\right)\right)}^{2}\right\}$$
where ${\tilde{R}}^{k}={\left[{\tilde{r}}_{1}^{k},{\tilde{r}}_{2}^{k},{\tilde{r}}_{3}^{k}\right]}^{T}$ denotes the optimized rotation matrix and ${\tilde{t}}^{k}={\left[{\tilde{t}}_{x}^{k},{\tilde{t}}_{y}^{k},{\tilde{t}}_{z}^{k}\right]}^{T}$ denotes the optimized translation vector. Through iteration, the optimized camera pose ${\tilde{R}}^{k}$ and ${\tilde{t}}^{k}$ can be obtained. This completes the initial pose estimation and optimization.

### 2.2. Two-Stage Iterative Depth Estimation Optimization

Although the optimized camera pose filters out the influence of image noise on pose estimation, it can not resolve the issue of missing depth information in monocular visual measurement. Therefore, using the optimized camera pose as the initial value, a two-stage iterative algorithm is employed to optimize the depth information. From the ideal pinhole camera model, rays emanating from the camera’s optical center as the model’s origin simultaneously traverse both the feature image point and the three-dimensional feature point. The unit ray vector $h_{i}^{k}$ can be expressed as(8)$$h_{i}^{k}=\frac{1}{\sqrt{{\left(u_{i}^{k}-c_{x}\right)}^{2}+{\left(v_{i}^{k}-c_{y}\right)}^{2}+f^{2}}}\left[\begin{array}{c} u_{i}^{k}-c_{x}\\ v_{i}^{k}-c_{y}\\ f \end{array}\right]$$

According to the principles of perspective projection, the coordinate of the feature point in the camera coordinate system, $P_{i}^{c}\left(k\right)={\left[X_{i}^{c}\left(k\right),Y_{i}^{c}\left(k\right),Z_{i}^{c}\left(k\right)\right]}^{T}$, and $P_{i}^{w}\left(k\right)$ should simultaneously lie on the ray unit vector $h_{i}^{k}$. Their coordinates in the camera coordinate system can be expressed as(9)$$P_{i}^{c}\left(k\right)=d_{i}^{k}h_{i}^{k}$$
where $d_{i}^{k}$ denotes the depth-of-field factor for feature point *i* during the *k*-th observation. By the collinearity principle, the collinearity equation yields(10)$$\begin{array}{cc} P_{i}^{c}\left(k\right) & ={\left(h_{i}^{k}\right)}^{T}P_{i}^{c}\left(k\right)h_{i}^{k}\\ \; & ={\left(h_{i}^{k}\right)}^{T}\left(R_{cw}^{k}P_{i}^{w}\left(k\right)+t_{cw}^{k}\right)h_{i}^{k} \end{array}$$
where $R_{cw}^{k}$ and $t_{cw}^{k}$ denote the rotation matrix and translation vector, respectively. Due to measurement errors between camera coordinates and true coordinates, the error function is formulated as(11)$$E\left(R_{cw}^{k},t_{cw}^{k}\right)=\sum_{i=1}^{n}{\left[\left(R_{cw}^{k}P_{i}^{w}\left(k\right)+t_{cw}^{k}\right)-{\left(h_{i}^{k}\right)}^{T}\left(R_{cw}^{k}P_{i}^{w}\left(k\right)+t_{cw}^{k}\right)h_{i}^{k}\right]}^{2}$$

Based on Equation (Equation 11), the rotation matrix $R_{cw}^{k}$ is initialized using ${\tilde{R}}^{k}$. After fixing the rotation matrix, the partial derivative with respect to the translation vector is taken to optimize the translation vector ${\tilde{t}}_{cw}^{k}$ and the depth of field $d_{cw}^{i}$(k):(12)$${\tilde{t}}_{cw}^{k}={\left(nI-\sum_{i=1}^{n}h_{i}^{k}{\left(h_{i}^{k}\right)}^{T}\right)}^{-1}\sum_{i=1}^{n}\left(h_{i}^{k}{\left(h_{i}^{k}\right)}^{T}-I\right)R_{cw}^{k}P_{i}^{w}\left(k\right)$$(13)$$d_{cw}^{i}\left(k\right)={\left(h_{i}^{k}\right)}^{T}\left(R_{cw}^{k}P_{i}^{w}\left(k\right)+{\tilde{t}}_{cw}^{k}\right)$$

Using the estimated depth of field, the point cloud in the camera coordinate system is then reconstructed, yielding the reconstructed camera point cloud coordinates ${\tilde{P}}_{i}^{c}\left(k\right)$:(14)$${\tilde{P}}_{i}^{c}\left(k\right)=d_{cw}^{i}\left(k\right)h_{i}^{k}$$

Given the optimized translation vector and depth, the translation vector and depth can be fixed while optimizing the rotation matrix. Due to the orthogonality of the rotation matrix, its optimization problem is transformed into a least-squares problem under orthogonal constraints. Therefore, the optimized rotation matrix ${\tilde{R}}_{cw}^{k}$ is solved using the Umeyama algorithm:(15)$${\tilde{R}}_{cw}^{k}=\underset{R_{cw}^{k}}{\arg\min}\sum_{i=1}^{n}{∥d_{cw}^{i}\left(k\right)h_{i}^{k}-\left(R_{cw}^{k}P_{i}^{w}\left(k\right)+{\tilde{t}}_{cw}^{k}\right)∥}^{2}$$

The error equation is constructed using the optimized pose and depth, and the maximum number of iterations is set to 100, with an iteration threshold of 0.1 mm:(16)$$E\left(R_{cw}^{k},t_{cw}^{k}\right)=\sum_{i=1}^{n}{∥d_{cw}^{i}\left(k\right)h_{i}^{k}-\left({\tilde{R}}_{cw}^{k}P_{i}^{w}\left(k\right)+{\tilde{t}}_{cw}^{k}\right)∥}^{2}$$

By alternately iterating Equations (12)–(16) until convergence, the high-precision depth and camera pose are obtained.

### 2.3. Reproducibility Error Propagation Model

As shown in Figure 3, the repeatability positioning accuracy of a robotic arm is typically defined as the dispersion of the end-effector’s pose after multiple repetitive motions. In essence, it calculates the dispersion of spatial discrete points, which can be quantified using the minimum enclosing sphere radius of the point set. When the robotic arm is equipped with a camera, since the end-effector and the camera are rigidly connected, their poses are related only by a fixed hand-eye transformation matrix. However, as repeatability positioning accuracy reflects the dispersion of poses, the camera pose can be used as a substitute for the end-effector pose in repeatability calculations. The eye-in-hand model is illustrated in Figure 4.

Current methods for monocular vision-based repeatability accuracy measurement predominantly rely on single-frame PnP solutions. These approaches statistically analyze pose distributions from multiple independent single-frame estimations to quantify robotic arm repeatability. However, due to the inherent scale ambiguity in monocular vision, these methods suffer from significantly compromised accuracy along the Z-axis. To address this fundamental limitation, we develop an error model of repetitive motion. The assumption that the robotic arm’s repeatability error follows a normal distribution is widely accepted in the field [24,25]. Based on this premise, the error equation is constructed as follows:(17)s(k)=t+e(k)
where s(k) represents the *k*-th observation, *t* denotes the spatial reference position that the repetitive motion attempts to reach (which remains constant regardless of the number of observations), and e(k) is the error of a single observation and follows a normal distribution. The spatial reference position t can be estimated by constructing the error equation. The above formulation indicates that the observational data consists of the reference position and the observational error. Now, based on the spatial reference position *t*, a new hypothesis is proposed: all observations are assumed to be taken at the spatial reference position *t*. Under this hypothesis, the observational data containing errors e(k) are backprojected to generate three-dimensional feature points, which inherently incorporate the observational errors e(k). This process integrates multi-frame information to transfer the repeatability error e(k) onto the point cloud, making the repeatability error e(k) explicitly quantifiable through point cloud registration.

### 2.4. Theoretical Pose Estimation

As the camera performs repetitive motion, under ideal conditions, the optimized camera pose ${\tilde{R}}_{cw}^{k}$ for each frame should be identical to ${\tilde{t}}_{cw}^{k}$. However, in practical operation, each observation by the robotic arm introduces random errors n(k) due to factors such as motor control fluctuations, minute variations in transmission clearance, and environmental noise. Therefore, a single observation value can be expressed as the sum of the observed value m(η) at the theoretical pose η and the error n(k):(18)$${\tilde{P}}_{i}^{img}\left(k\right)=m\left(\eta \right)+n\left(k\right)$$(19)$$\left\{\begin{array}{c} {\tilde{P}}_{i}^{img}\left(k\right)=\pi \left({\tilde{R}}_{cw}^{k},{\tilde{t}}_{cw}^{k},P_{i}^{w}\left(k\right)\right)\\ m\left(\eta \right)=\pi (\eta ,P_{i}^{w}\left(k\right)) \end{array}\right.$$
where ${\tilde{P}}_{i}^{img}\left(k\right)$ denotes the *k*-th observation value; n(k) represents the *k*-th error, which follows a Gaussian distribution with mean zero and variance $\delta^{2}$. The errors from each observation are mutually independent. The theoretical pose η can be expressed as(20)$$\left\{\begin{array}{c} \eta =[R_{true}|t_{true}]\\ R_{true}={\left[R_{1},R_{2},R_{3}\right]}^{T}\\ t_{true}={\left[T_{x},T_{y},T_{z}\right]}^{T} \end{array}\right.$$
where the theoretical pose η comprises a rotational component $R_{true}$ and a translational component $t_{true}$. Given that the theoretical pose η remains constant over *K* repetitive motions, while errors n(k) arise from random factors, the *K* observations can be modeled as a random signal: a single observation value can be expressed as the sum of the observed value m(η) at the theoretical pose η and the error n(k). Since error n(k) follows a Gaussian distribution, the maximum likelihood method can be employed to estimate the theoretical pose η. Considering the Gaussian nature of the error, maximizing the likelihood function is equivalent to minimizing the sum of squares of the reprojection errors between all observed values and the theoretical true values m(η) [26]. Therefore, the global optimization problem for determining the theoretical pose η can be formulated as(21)$$\begin{array}{cc} \eta & =\underset{\eta }{\arg\min}\sum_{k=1}^{K}{∥{\tilde{P}}_{i}^{img}\left(k\right)-m\left(\eta \right)∥}^{2}\\ \; & =\sum_{k=1}^{K}{\left[{\tilde{P}}_{i}^{img}\left(k\right)-m\left(\eta \right)\right]}^{T}\left[{\tilde{P}}_{i}^{img}\left(k\right)-m\left(\eta \right)\right] \end{array}$$
where $∥{\tilde{P}}_{i}^{img}\left(k\right)-m\left(\eta \right)∥$ denotes the Euclidean distance between the *k*-th observation ${\tilde{P}}_{i}^{img}\left(k\right)$ and the theoretical true value m(η). Solving this nonlinear least-squares problem yields the theoretical pose η. This process effectively fuses information from *K* observations, mitigates the impact of random errors in individual observations, and estimates the theoretical pose η via maximum likelihood estimation.

### 2.5. Cramér–Rao Bound

For parameter estimation problems, the choice of estimation criterion directly determines the performance of the estimate. To understand the potential performance of parameter estimation, the Cramér–Rao bound (CRB) is adopted as the evaluation benchmark. According to optimal estimation theory, the CRB represents the lower variance limit for all unbiased estimators of a parameter and can be expressed as(22)$$C\left(\eta \right)\geq C_{CR}\left(\eta \right)$$

In the equation, C(η) represents the variance of the parameter estimation error, and $C_{CR}\left(\eta \right)$ denotes the Cramér–Rao bound, which can be obtained by inverting the Fisher information matrix J:(23)$$C_{CR}\left(\eta \right)=J^{-1}$$

Based on the aforementioned stochastic signal model, the normalized likelihood function can be derived according to maximum likelihood theory:(24)$$L\left(\eta \right)=-ln\left[\det\left(Q\right)\right]-\frac{1}{K}\sum_{k=1}^{K}\left\{{\left[{\tilde{P}}_{i}^{img}\left(k\right)-m\left(\eta \right)\right]}^{T}Q^{-1}\left[{\tilde{P}}_{i}^{img}\left(k\right)-m\left(\eta \right)\right]\right\}$$(25)$$Q=\delta^{2}I$$
where det(·) denotes the matrix determinant and *Q* is the covariance matrix. Based on the likelihood function derived from the above equation, the general form of the Fisher information matrix can be obtained as(26)$$\begin{array}{cc} {\left[J\right]}_{i,j} & =E[\frac{\partial L(\eta )}{\partial \eta_{i}}\;\;\frac{\partial L(\eta )}{\partial \eta_{j}}]\\ \; & =K\left\{tr\left[Q^{-1}\frac{\partial Q}{\partial \eta_{i}}Q^{-1}\frac{\partial Q}{\partial \eta_{j}}\right]\right\}+2K\left[\frac{\partial m(\eta )}{\partial \eta_{i}}Q^{-1}\frac{\partial m(\eta )}{\partial \eta_{j}}\right] \end{array}$$

In the pose estimation problem, since the first term in the above equation is independent of the parameters to be estimated, and the noise is assumed to be isotropic Gaussian, with *Q* being the covariance matrix of the random variables, it follows that the Fisher information matrix for the pose estimation problem can be expressed as(27)$${\left[J\right]}_{i,j}=\frac{2K}{\delta^{2}}\left[\frac{\partial m^{T}\left(\eta \right)}{\partial \eta_{i}}\frac{\partial m(\eta )}{\partial \eta_{j}}\right]$$

Consequently, the Cramér–Rao bound of the parameters to be estimated can be expressed as(28)$$C_{CR}\left(\eta \right)=\frac{\delta^{2}}{2K}{\left[\frac{\partial m^{T}\left(\eta \right)}{\partial \eta_{i}}\frac{\partial m(\eta )}{\partial \eta_{j}}\right]}^{-1}$$

As can be seen from the above equation, the Cramér–Rao bound for parameter estimation decreases linearly with the increase in the number of observations and increases linearly with the increase in noise power.

### 2.6. Repeatability Accuracy Calculation

In existing methods for calculating repeatability accuracy using monocular vision, the PnP algorithm is typically employed for camera pose estimation and optimization, with repeatability accuracy calculated directly from the optimized pose. Although the PnP algorithm can directly solve for poses using 2D–3D matched point pairs, the ICP algorithm exhibits lower estimation uncertainty and achieves more precise pose estimation when depth value information is precisely known [27]. Therefore, this paper introduces the ICP algorithm for calculating repeatability accuracy.

As the ICP is a point cloud registration algorithm, only the pose of each observation and the theoretical pose are currently known; therefore, a new observation model must be constructed to incorporate the ICP. This leads to the following hypothesis: assuming the poses of *K* observations are all theoretical poses η, if the 2D image point coordinates remain unchanged, the calibration board must be moved to ensure that the relative pose remains constant. Based on the principle of constant relative pose, it follows that the depth information will not change. Consequently, by backprojecting the observations into three-dimensional space, the coordinates of the three-dimensional feature points on the repositioned calibration plate can be obtained, yielding the corresponding point pairs. The coordinate pairs ${\tilde{P}}_{i}^{w}\left(k\right)$ and $P_{i}^{w}\left(k\right)$ form corresponding point pairs:(29)$${\tilde{P}}_{i}^{w}\left(k\right)=\pi^{-1}\left({\tilde{P}}_{i}^{img}\left(k\right),d_{cw}^{i}\left(k\right),\eta \right)$$
where $\pi^{-1}\left({\tilde{P}}_{i}^{img}\left(k\right),d_{cw}^{i}\left(k\right),\eta \right)$ is the backprojection function for the ideal pinhole camera model. This transfers the robotic arm error from the arm’s pose to the calibration plate’s pose. The robotic arm error n(k) can then be quantified by registering the corresponding point pairs using the point-to-point ICP algorithm:(30)$$n\left(k\right)=\left[\Delta R^{k},\Delta t^{k}\right]=\underset{\Delta R^{k},\Delta t^{k}}{\arg\min}\sum_{i=1}^{n}{∥\left(\Delta R^{k}{\tilde{P}}_{i}^{w}\left(k\right)+\Delta t^{k}\right)-P_{i}^{w}\left(k\right)∥}^{2}$$
where $\Delta R^{k}$ denotes rotational deviation and $\Delta t^{k}$ denotes translational deviation. The maximum number of iterations and the convergence threshold for the ICP algorithm are set to $10^{5}$ and $10^{-8}$. This transformation converts the problem of solving the pose of a stationary target PnP through multiple observations into solving the relative pose of a slightly moving target through multiple ICP-matched point clouds. If the translational deviation component $\Delta t^{k}$ obtained from each ICP iteration is regarded as a discrete point in three-dimensional space, after *K* repetitions of the motion, the set of translational deviation points can be obtained:(31)$$\tau =\{\Delta t^{1},\Delta t^{2},\ldots ,\Delta t^{K}\}$$

By determining the radius of the minimum enveloping sphere for this point set, the repeatability accuracy of the robotic arm can be quantified.

## 3. Results

The experimental setup comprised a robotic arm, an industrial camera, and a circular calibration plate. The robotic arm was an Elfin-Pro E03-Pro collaborative robot featuring six axes of motion and weighing 18 kg, with a maximum payload of 3 kg and a working range of 590 mm. According to the manufacturer’s datasheet, it has a repeatability positioning accuracy of ±0.02 mm. All experiments were conducted in a temperature-controlled environment (23 ± 2 °C) to ensure that the robot reached thermal equilibrium, thereby minimizing potential accuracy drift due to temperature variations. The camera was an SVersion-HJ-RGBD-90 industrial camera with a resolution of 3120 × 4096 pixels. The calibration board was a CGB-020 5 × 4 dot calibration board, as illustrated in Figure 5. Our computer was configured with an Intel Core i7-12700K CPU and an NVIDIA GeForce RTX 3060 GPU. All image data in the experiments use pixels as the unit, while all vector data are expressed in millimeters (mm).

First, the algorithm’s efficacy was validated. The algorithm validation experiment primarily assessed initial pose optimization and theoretical pose estimation. This paper employed weighted maximum likelihood to optimize the initial pose. To validate the effectiveness of the improved IGG 3 function weight-allocation strategy, optimization was conducted using both maximum likelihood (ML) and weighted maximum likelihood (WML) under identical initial pose conditions. Performance was assessed by statistically evaluating the reprojection error, with the results presented below in Table 1.

The experimental results demonstrate that incorporating the improved IGG 3 function weight-allocation strategy effectively reduced the reprojection error. Specifically, the maximum error of WML decreased by 87.4% compared to ML, proving that the modified IGG 3 function dynamically adjusted weights to suppress outlier effects and mitigate the impact of inherent salt-and-pepper noise on pose estimation outcomes. The mean error and standard deviation of WML improved by 44.11% and 76.69%, respectively, compared to ML. This indicates that the introduced IGG 3 function weight-allocation strategy enables the optimization objective to focus more on high-confidence matching points, significantly enhancing the accuracy and consistency of pose estimation and demonstrating that the algorithm delivers stable gains under normal operating conditions while maintaining robustness in extreme scenarios.

This paper constructed a random signal model based on the statistical characteristics of errors and employed the maximum likelihood method to estimate theoretical poses. To validate the effectiveness of the maximum likelihood estimated theoretical poses and assess whether the estimation results can serve as the system’s theoretical poses, the theoretical boundary was calculated. The proposed method was compared with the Cramér–Rao bound using the mean reprojection error as the evaluation metric. The results are shown in Table 2 and Figure 6.

Experimental results demonstrate that across five independent replicate experiments, the accuracy of the maximum likelihood global optimization algorithm approached that of the Cramér–Rao bound. The maximum difference between this method and the Cramér–Rao bound was 0.0267 pixels, with the minimum difference being 0.0086 pixels. Both differences remained within 0.03 pixels, proving that the maximum likelihood method based on the random signal model can effectively estimate theoretical poses.

Second, simulation experiments analyzed factors influencing the pose estimation module algorithm. As the initial pose estimation employed the PnP algorithm, to identify key factors affecting the initial pose optimization algorithm, the proposed method was compared with the classical PnP improved algorithm [10,11,12,13,14] and the orthogonal iteration algorithm [19]. In the experiments, a scenario with known camera intrinsic parameters was simulated. The synthetic virtual camera resolution was assumed to be 640 × 480 pixels with a fixed focal length of 800 pixels. Three-dimensional feature points were randomly generated, distributed on a plane with coordinates ranging from [−2, 2] m × [−2, 2] m. Two-dimensional feature points were projected using an ideal pinhole camera model. Employing a controlled variable method, the number of reference points N and noise level ω were analyzed as study variables. For each variable value, 500 simulation experiments were conducted. Algorithm performance was evaluated by calculating the mean and median of the rotational and translational errors. The variable parameter settings are detailed in Table 3, with the experimental results presented in Figure 7 and Figure 8.

The experimental results demonstrate that the error of all algorithms decreased as the number of reference points increased, while it rose with increasing noise levels. Furthermore, at a fixed noise level, accuracy gradually converged once the number of reference points reached a certain threshold. As EPnP is a non-iterative linear algorithm, it is sensitive to noise, making it difficult to guarantee accuracy when few feature points are available. DLS disregards geometric constraints, making it prone to singularities during computation and resulting in overall instability. Although OI is an iterative method, its accuracy and stability deteriorated when reference points were scarce or near singularity, frequently trapping it in local optima. The RPnP, OPnP, and SRPnP algorithms all exhibited high precision; however, as RPnP is a suboptimal algorithm, its accuracy was marginally lower than that of OPnP and SRPnP. In contrast, the proposed method enhances accuracy by using a weighted maximum likelihood approach to optimize initial poses, improving the extraction precision of two-dimensional feature points. It constructs a random signal model and employs maximum likelihood estimation for theoretical pose estimation, thereby preventing estimation results from becoming trapped in local optima. Consequently, it outperformed the comparison methods in both accuracy and stability.

Finally, a comparative analysis was conducted. This analysis primarily encompassed module comparisons and comparisons with similar methods. Within the robotic arm’s workspace, an industrial camera mounted on the robotic arm captured 150 calibration plate images. A portion of the dataset images is shown in Figure 9.

The above images represent calibration plate photographs captured under five different poses: P1–P5. To quantitatively evaluate the impact of each module’s algorithm on the final repeatability measurement accuracy, a comparative experiment was designed using 30 datasets from the P1 pose. The comparison schemes were as follows:Scheme A: Basic PnP (unoptimized);Scheme B: A + WML (initial optimization);Scheme C: B + two-stage iteration + ML (global optimization);Scheme D: C + ICP (proposed method).

Scheme A performs the PnP solution directly after data acquisition, using the first frame’s pose as the baseline to compute the relative positional relationships for subsequent frames, thereby determining repeatability accuracy. Scheme B optimizes the PnP solution from Scheme A before calculating relative positions and determining repeatability accuracy. Scheme C builds upon Scheme B by employing a two-stage iterative algorithm to optimize depth and pose. It uses a maximum likelihood method based on a random signal model to estimate theoretical poses. Using these theoretical poses as reference, it calculates each frame’s relative position to the theoretical pose to determine repeatability accuracy. Scheme D represents the method proposed in this paper. Building upon Scheme C, it constructs a three-dimensional homographic point cloud based on the theoretical pose and the optimized depth field. This enables the introduction of the ICP algorithm to compute repeatability accuracy. The specific comparative experimental results are presented in Table 4.

The comparative experimental results demonstrated that the proposed method achieved repeatability accuracy of 0.0115 mm, significantly outperforming alternative approaches. The specific analysis was as follows: Scheme A exhibited repeatability accuracy of 17.9465 mm, markedly higher than that of other approaches. This is primarily because single-frame PnP solutions are susceptible to observational noise interference, leading to substantial positional estimation deviations and poor stability. If the inaccurate single-frame pose from the initial frame is directly used as the reference for calculating relative poses, the relative poses incorporate gross errors, thereby causing repeatability accuracy measurements to fail. Following the introduction of WML optimization in Scheme B, repeatability accuracy improved by 93%. Although this scheme effectively mitigated the impact of gross errors on the results by modeling the statistical characteristics of observational noise, thereby significantly enhancing single-frame pose stability, it failed to resolve the inaccuracy in the depth calculation under the coplanar configuration of monocular visual features. This compromised the accuracy of the three-dimensional repeatability results. Scheme C builds upon Scheme B by employing a two-stage iterative algorithm. It optimizes depth and relative pose based on collinearity constraints and estimates the theoretical pose via a maximum likelihood method grounded in a random signal model. This resolved depth estimation inaccuracies and enhanced repeatability accuracy. However, PnP methods reliant solely on 2D–3D matching face inherent limitations. The proposed method incorporates the ICP algorithm into Scheme C, achieving significantly higher accuracy than that scheme. This demonstrates that the ICP registration scheme for homonymous point clouds—based on the theoretical pose η and the optimized depth field backprojection—effectively overcomes the projection error sensitivity inherent in 2D–3D matching, yielding more precise repeatability accuracy.

To demonstrate the superiority of the proposed method, we conducted a comparative analysis against existing monocular vision-based repeatability accuracy calculation methods. Given identical input image data, we compared the repeatability accuracy results of our method with those from [16,18]. The specific repeatability accuracy calculations and corresponding error analysis are presented below.

Through the analysis of the data in Table 5 and Table 6, the method proposed in this paper significantly outperformed the comparative methods in repetitive positioning accuracy measurement, with measurement values within 0.0175 mm and a mean repeatability positioning accuracy of 0.0128 mm, which is on the same order of magnitude as the factory-calibrated accuracy (±0.02 mm) of the test robot, indicating that the proposed method has minimal measurement error and is capable of accurately capturing the true performance of the robotic arm. Further analysis of error statistics indicates that the proposed method surpassed the comparative methods in both mean error and standard deviation metrics, demonstrating small error fluctuations in the measurement results and no significant outlier error points. Specifically, the method in [16], based on the PnP algorithm, is sensitive to feature point distribution and calculates repetitive positioning accuracy through single-frame pose estimation results. Although it optimizes the extracted 2D feature points during image point extraction to avoid the influence of gross errors, resulting in relatively good stability, it does not consider the impact of depth estimation errors on the results. Since this algorithm is similar to Scheme B, its accuracy can only be controlled at the millimeter level. While the PnP + OI combined method proposed in [18] improves overall accuracy through nonlinear optimization, it relies on 2D–3D matching-based PnP methods that possess inherent limitations. In this paper, an improved IGG3 weighting function is adopted to suppress feature point outliers, a two-stage iterative strategy is employed to optimize depth and pose, and a maximum likelihood method based on a random signal model is constructed to estimate the theoretical pose. The theoretical pose is then used to backproject 3D points as observations, combined with the ICP algorithm to break through the accuracy limits of 2D–3D matching. The introduction of the ICP algorithm indeed increased computational cost by an average of 8 s, as reported in Table 6. This trade-off is justified through an analysis of algorithmic complexity and the fundamental objectives of this work. The complexity of the ICP algorithm is dominated by the nearest-neighbor search. Given point clouds of size N and M, the per-iteration complexity of ICP using a KD-Tree is O(NlogM), leading to an overall complexity of O(kNlogM) for k iterations, which is a manageable polynomial-time complexity. Considering that the primary objective of this study is high-accuracy 3D repeatability assessment (as opposed to high-speed online control), and that traditional monocular methods suffer from inherent depth ambiguity, the fixed computational cost is warranted. The ICP registration elevates the measurement from the 2D image space to a metric 3D space, fundamentally resolving the scale ambiguity issue. This is considered a critical advancement for measurement tasks where precision is paramount.

## 4. Conclusions

This paper addresses critical issues in monocular vision-based robotic arm repeatability measurement, including missing statistical assumptions in observational data, non-global optimization of pose estimation, and insufficient depth estimation accuracy. It proposes a quantitative method based on likelihood pose estimation. By optimizing single-frame poses from PnP estimates, constructing a random signal model, and employing the approach to fuse multi-frame observations for theoretical pose estimation, this approach overcomes the accuracy limitations of traditional 2D–3D matching through optimized depth information and ICP algorithms. Experimental validation demonstrates that the proposed method achieves repeatability accuracy consistently within 0.0068–0.0175 mm, with a standard deviation of 0.0038 mm. This represents improvements of 85% and 87% over comparative methods, offering enhanced precision and stability. It provides novel theoretical and methodological pathways for applying visual measurement techniques in robotic arm performance evaluation. Although the proposed method performs well in controlled environments with stable lighting and no occlusions, variable illumination and partial occlusion in industrial settings remain critical practical challenges. In future work, we will focus on enhancing the robustness of the approach. Specific directions include investigating illumination-invariant feature descriptors and developing robust estimation algorithms capable of identifying and rejecting outlier observations caused by occlusions. The ultimate goal is to adapt this measurement system for deployment in more complex industrial field environments.

## Figures and Tables

**Figure 1 sensors-25-07089-f001:**
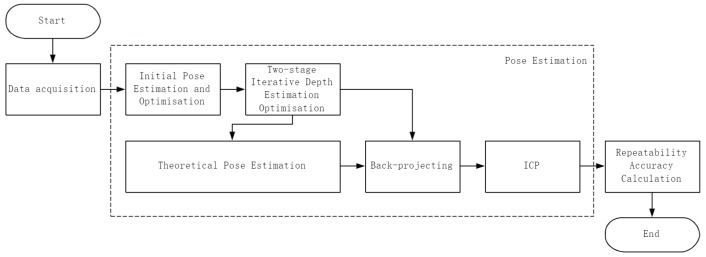
Algorithm flowchart.

**Figure 2 sensors-25-07089-f002:**
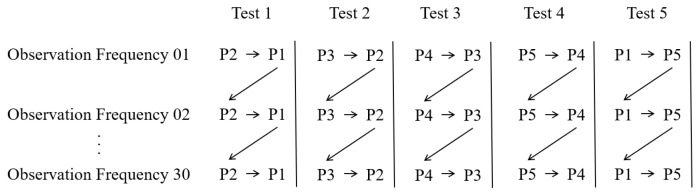
Image acquisition flowchart.

**Figure 3 sensors-25-07089-f003:**
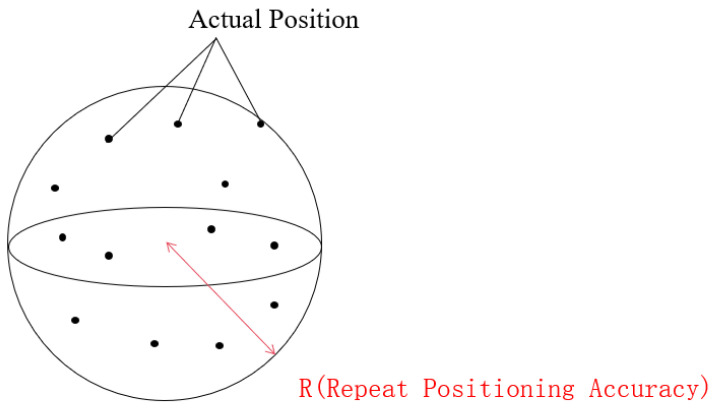
Repeatability positioning accuracy.

**Figure 4 sensors-25-07089-f004:**
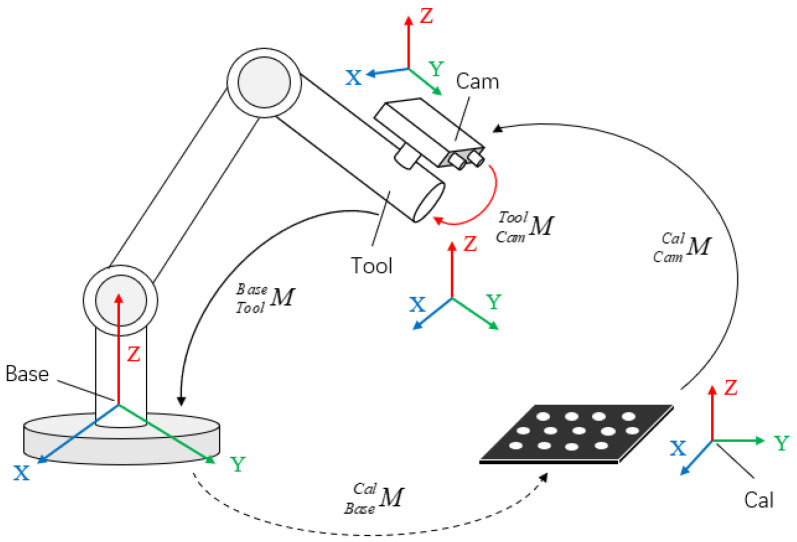
Eye-in-hand model.

**Figure 5 sensors-25-07089-f005:**
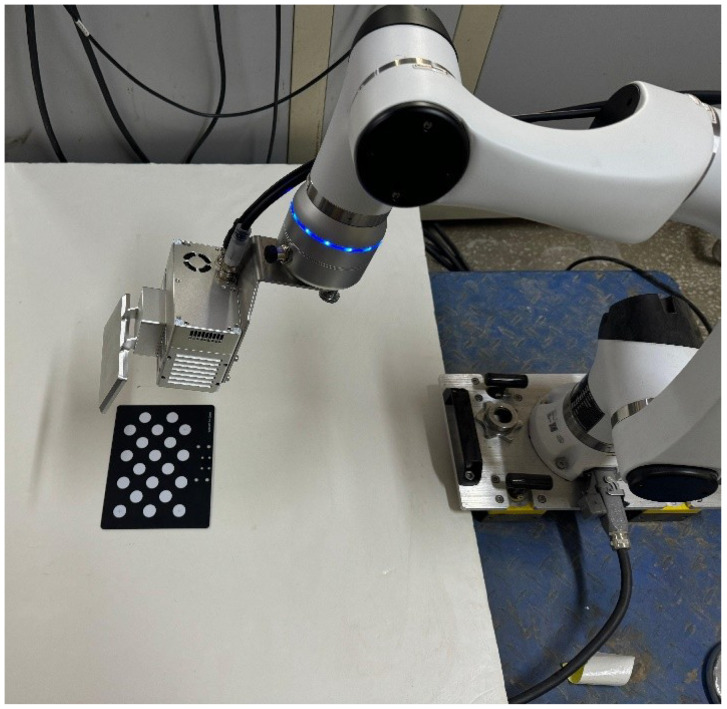
Experimental setup.

**Figure 6 sensors-25-07089-f006:**
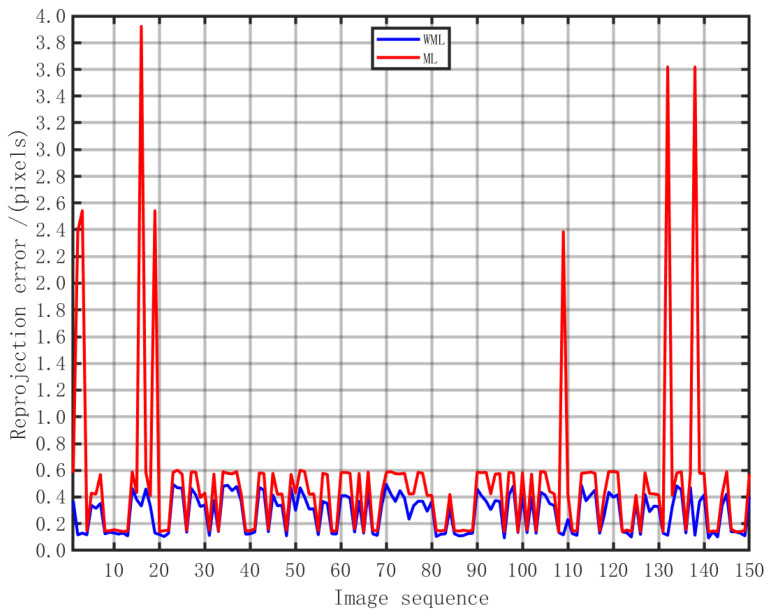
Error comparison chart.

**Figure 7 sensors-25-07089-f007:**
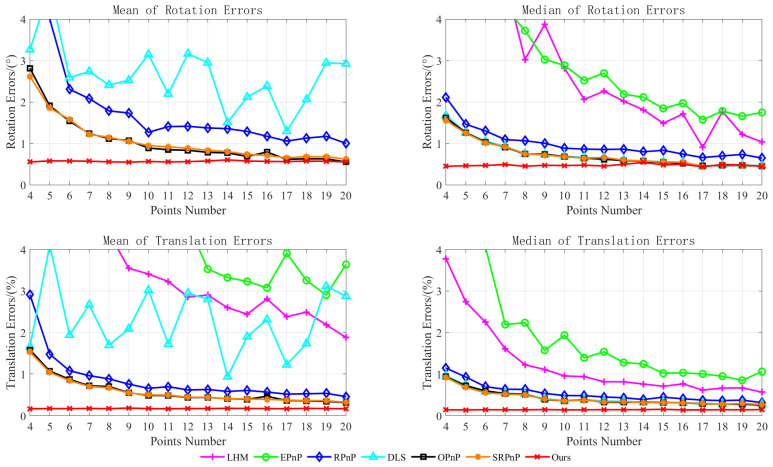
Variations in reference point count.

**Figure 8 sensors-25-07089-f008:**
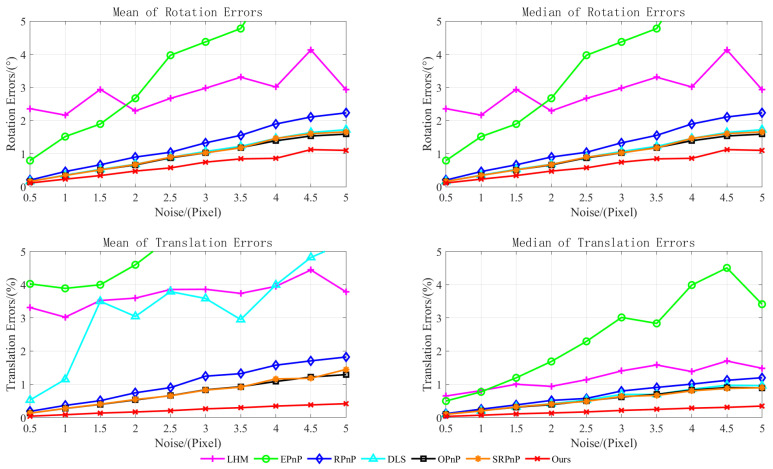
Variations in noise level.

**Figure 9 sensors-25-07089-f009:**
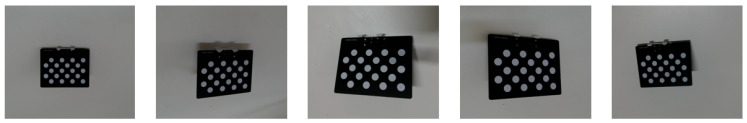
Calibration plate image data.

**Table 1 sensors-25-07089-t001:** Effectiveness analysis results.

	Minimum/Pixels	Maximum/Pixels	Mean/Pixels	Standard Deviation/Pixels
WML	0.0937	0.4940	0.2835	0.1402
ML	0.1395	3.9224	0.5072	0.6016

**Table 2 sensors-25-07089-t002:** Performance evaluation findings.

	1/Pixels	2/Pixels	3/Pixels	4/Pixels	5/Pixels
Ours	0.0383	0.0363	0.0417	0.0473	0.0432
Cramér–Rao bound	0.0116	0.0277	0.0346	0.0336	0.0193

**Table 3 sensors-25-07089-t003:** Variable parameter settings.

	Minimum	Maximum	Step
N	4	20	1
ω/pixels	0	5	0.5

**Table 4 sensors-25-07089-t004:** Module comparative analysis results.

	A	B	C	D
Repeatability/mm	17.9465	1.2730	0.9792	0.0115

**Table 5 sensors-25-07089-t005:** Repeatability positioning accuracy calculation results.

	P1/mm	P2/mm	P3/mm	P4/mm	P5/mm
[16]	1.0569	1.0898	1.0303	1.0702	1.1041
[18]	0.8049	0.8088	0.7567	0.8107	0.8331
Our method	0.0115	0.0121	0.0068	0.0162	0.0175

**Table 6 sensors-25-07089-t006:** Error and computational efficiency analysis.

	Mean/mm	Standard Deviation/mm	Time Cost/s
[16]	1.0703	0.0257	3.3454
[18]	0.8028	0.0286	7.1082
Our method	0.0128	0.0038	11.1390

## Data Availability

The original contributions presented in the study are included in the article; further inquiries can be directed to the corresponding author.

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
