# Peer review of "A Likelihood-Based Pose Estimation Method for Robotic Arm Repeatability Measurement Using Monocular Vision"

_sensors, 2025, doi:10.3390/s25227089_

Round 1

Reviewer 1 Report

Comments and Suggestions for Authors

In light of the limitations of existing monocular vision-based methods for measuring robotic arm repeatability, this study proposes a quantitative method for evaluating repeatability accuracy based on likelihood pose estimation. The results demonstrate that the proposed method effectively improves measurement accuracy and holds practical value for industrial applications. The main comments are as follows:

  1. The model description is unclear regarding which components are newly proposed and which are derived from existing literature.

  2. While the abstract highlights "an average accuracy improvement of 1.0575 mm," no quantitative comparative advantages are provided in the model comparison. In which specific engineering applications can the reported accuracy enhancement be applied?

  3. Some figures and tables are unclear and require revision.

  4. The results analysis should specify the parameter values used in both the proposed model and the comparative models to facilitate reproducibility.

Reviewer 2 Report

Comments and Suggestions for Authors

General and Structural Comments

  1. Title clarity: The title conveys the method’s name but does not specify the domain (robotic arms under monocular vision). Adding this context would enhance discoverability.
  2. Abstract conciseness: The abstract is overly dense; it introduces too many technical terms (e.g., “weighted maximum likelihood,” “stochastic signal model”) without briefly defining them. Simplifying for clarity would help general readers.
  3. Lack of numerical summary in abstract: Although accuracy improvement is mentioned (1.0575 mm), there is no reference to statistical measures (variance, number of samples). Include these to strengthen credibility.
  4. Missing DOI and submission metadata: The manuscript still shows placeholder metadata (e.g., “Journal Not Specified”). This should be updated before final submission.
  5. Literature review completeness: The introduction references well-known PnP algorithms but omits recent developments in learning-based or hybrid optimization methods (e.g., deep learning pose estimators).
  6. Inconsistent reference formatting: Some citations (e.g., Luo et al., line 54) have formatting errors such as “[?, luo2020endoscopic].” References must be checked for completeness and correctness.
  7. Innovation claim: The proposed “likelihood pose method” is described as novel, but similar multi-frame maximum likelihood fusion approaches exist in structure-from-motion and photogrammetry. Clarify the specific novelty.

Technical and Methodological Comments

  1. Mathematical rigor of likelihood function (Eq. 4): The derivation assumes Gaussian i.i.d. errors but does not justify why this distribution is appropriate for visual feature noise. Empirical noise modeling could support this assumption.
  2. Weighting function details: The “improved IGG 3 function” is mentioned but not mathematically defined. Provide its explicit form and parameters to ensure reproducibility.
  3. Noise model validation: The manuscript assumes zero-mean Gaussian noise, yet industrial cameras often show anisotropic or spatially correlated noise. Discuss the robustness of the method under non-Gaussian noise.
  4. Equation numbering consistency: Ensure all equations are consecutively numbered; some intermediate equations (e.g., between Eq. 8–12) lack contextual explanation.
  5. Camera calibration parameters: The calibration method for intrinsic parameters (fx, fy, cx, cy) is briefly mentioned but lacks reference to calibration uncertainty or reprojection residuals. Include these to quantify input precision.
  6. Depth estimation model: Equation (9) minimizes an error function without specifying convergence criteria or stopping tolerance for the two-stage iterative algorithm. Add convergence analysis.
  7. Umeyama algorithm justification: It is unclear why the Umeyama method (Eq. 12) was selected over Horn’s quaternion-based solution or SVD-based alignment. Compare methods or justify the choice.
  8. Theoretical pose estimation (Eq. 16): The likelihood fusion assumes constant intrinsic and extrinsic parameters across K frames. Discuss implications if small camera drift or lens distortion occurs.
  9. ICP integration: The manuscript introduces ICP in the final stage but does not define which variant (point-to-point, point-to-plane). The variant and parameters (max iterations, convergence ε) must be stated.
  10. Outlier rejection in ICP: No mention is made of correspondence filtering (e.g., trimming, RANSAC, or rejection thresholds). This is essential for repeatability accuracy robustness.
  11. Complexity analysis: The ICP addition increases time cost by ~8 s (Table 6). Provide an algorithmic complexity estimation (O(n), O(n²)) to justify this trade-off.
  12. Repeatability metric definition: The “minimum enveloping sphere radius” is innovative but unusual. Cite ISO 9283 definitions and clarify how this measure corresponds to standard robot repeatability metrics.
  13. Error propagation: There is no analysis of how uncertainty from pose estimation and depth reconstruction propagates into final repeatability accuracy. Add error propagation equations or Monte Carlo validation.
  14. Cramér–Rao bound comparison: While mentioned, the derivation of the CRB for this system is missing. Include a theoretical derivation or simulation to validate proximity to the bound.
  15. Robotic setup reproducibility: The paper lists the robot and camera models but omits key parameters such as robot joint resolution, temperature stability, or calibration board reflectance—all of which affect accuracy.
  16. Lighting and environmental factors: The method assumes consistent illumination but does not test performance under variable lighting or partial occlusion, which are critical in industrial settings.
  17. Number of repetitions: Each pose is repeated 30 times, but justification for this sample size (statistical sufficiency for Gaussian modeling) is not given. Include variance convergence analysis.
  18. Coordinate transformations: Equation (17) introduces π⁻¹ for backprojection, but the transformation pipeline (camera → world → robot) is not diagrammed. Add a coordinate transformation schema.

Experimental Design and Results

  1. Baseline method selection: Only Paper [15] is used for quantitative comparison. Include more recent benchmarks (e.g., deep-PnP, ICP-Refined EPnP, or hybrid SLAM-based calibration) for a fairer evaluation.
  2. Statistical significance: Differences of ~0.01 mm are reported, but no standard deviation or statistical significance testing (e.g., t-test) is performed to confirm meaningful improvement.
  3. Noise simulation fidelity: In simulated experiments (Fig. 6), the pixel noise model ω is varied, but the mapping between ω and real sensor SNR is missing. Clarify how this parameter corresponds to physical conditions.
  4. Visualization quality: Figures (e.g., Fig. 3–7) lack scale bars, axis labels, and color legends. Add these to make visual results interpretable.
  5. Data normalization: It is unclear whether all datasets were normalized in units (mm vs pixels) before computing errors. Explicitly state the unit consistency for reproducibility.
  6. Runtime measurement: The reported time cost (11.139 s) lacks details on computational hardware (CPU, GPU). Include specifications to allow replication of performance metrics.
  7. Comparison fairness: The proposed method includes additional steps (ICP, ML), whereas the baseline omits these. Make clear that comparison is between complete pipelines with equal input data.

Discussion and Conclusions

  1. Overstatement of novelty: While performance gains are strong, the method builds heavily upon classical maximum likelihood and ICP frameworks. Temper the claim of “new theoretical pathway” to “enhanced integration framework.”
  2. Industrial applicability: The authors mention “applicable in actual production,” yet no real industrial case study is provided. Add a demonstration in a real robotic assembly or welding setup.
  3. Scalability to dynamic environments: The method assumes static calibration. Discuss adaptability to moving cameras or multi-robot collaborative systems.
  4. Future work direction: The conclusion suggests outdoor extensions but does not outline specific technical approaches (e.g., adaptive exposure control, light-invariant descriptors). Adding these would strengthen research continuity.

Round 2

Reviewer 1 Report

Comments and Suggestions for Authors

In the revised manuscript, the authors have addressed all comments in detail, and the paper is now recommended for acceptance.

Reviewer 2 Report

Comments and Suggestions for Authors

The authors have adequately addressed all the required revisions. Therefore, the acceptance of the manuscript in its current form is recommended.